# LiftQuant: Continuous Bit-Width LLM via Dimensional Lifting and Projection

**Liulu He** [1]   **XuanAng Liu** [1]   **Juntao Liu** [2]   **Taolue Feng** [3]   **Ting Lu** [3]   **Chunsheng Gan** [3]   **Zhiyv Peng** [1]   **Yuan Du** [1]
**Huanrui Yang** [4] [*]   **Yijiang Liu** [5] [*]   **Li Du** [1] [*]

## Abstract

Existing quantization methods are fundamentally limited by rigid, integer-based bit-widths (e.g., 2, 3-bit), resulting in a "deployment gap" where Large Language Models cannot be optimally fitted to specific memory budgets. To bridge this gap, we introduce LiftQuant, a novel framework that enables continuous bit-width control for true Pareto-optimal deployment. The core innovation is a "lift-then-project" mechanism which approximates low-dimensional weight vectors by projecting a simple 1-bit lattice from a higher-dimensional "lifted" space. Crucially, the effective bit-width is determined simply by the ratio of the lifted dimension to the original dimension, which allows the bit-width to be tuned quasi-continuous as the dimension is a flexible structural parameter. This projection generates a structured yet non-uniform codebook, capturing the expressive power of Vector Quantization (VQ). While beneficial over VQ, LiftQuant's decoding path relies solely on linear transformations and 1-bit uniform quantizers, retaining hardware-friendly nature. This flexibility is transformative: LiftQuant enables a 70B LLM to be compressed to 2.4 bits to precisely fit a 24GB GPU, where its performance significantly surpasses state-of-the-art 2-bit models fitted on the same device. Our code and ckpt is available at https://github.com/Heliulu/LiftQuant.

---

[*]Co-corresponding Authors. This work was supported in part by the National Key Research and Development Program of China under Grant 2022YFB4400900, in part by the Natural Science Foundation of China under Grants 62371223.   [1]Nanjing University, China [2]China Mobile Research Institute, China [3]Xiaomi Corporation, China [4]University of Arizona, USA [5]Nanjing University of Information Science and Technology, China. Correspondence to: Huanrui Yang <huanruiyang@arizona.edu>, Yijiang Liu <liuyijiang@nuist.edu.cn>, Li Du <ldu@nju.edu.cn>.

## 1. Introduction

Large Language Models (LLMs) have demonstrated unprecedented capabilities across a wide range of tasks, but their massive parameter counts pose a severe challenge for deployment. The "memory wall" remains the primary bottleneck: running state-of-the-art models (e.g., 70B parameters) typically requires high-end, multi-GPU clusters, making them inaccessible for deployment on commodity hardware or edge devices. Consequently, weight-only quantization has emerged as a standard practice to compress these models into manageable footprints.

However, a fundamental inefficiency plagues current quantization paradigms: the rigidity of integer bit-widths. Existing methods, whether based on Uniform Quantization (UQ) or Vector Quantization (VQ), force users to choose between discrete compression levels (e.g., 2-bit, 3-bit, or 4-bit). This creates a significant "deployment gap" between model size and hardware capacity. For instance, consider deploying a Llama-3-70B model on a consumer-grade GPU with 24GB of VRAM, a 3-bit quantization is too large to fit, while a 2-bit quantization, though small enough, suffers from a catastrophic drop in reasoning capability. The hardware's memory capacity between 2-bit and 3-bit is effectively wasted, and the model's potential performance is capped by the coarse granularity of the quantization scheme.

To bridge this gap, we introduce LiftQuant, a novel quantization framework that transforms the rigid selection of bit-widths into a continuous design space. LiftQuant is, to the best of our knowledge, the first framework to enable arbitrary fractional bit-widths (e.g., 2.4-bit) for LLMs, allowing for true Pareto-optimal deployment under strict memory constraints. Our approach departs from traditional scalar or vector codebook learning. Instead, we employ a "lift-then-project" mechanism: we construct each weight vector as a learned linear combination of elements from a simple 1-bit lattice defined in a higher-dimensional space. This approach effectively decouples the quantization rate from the coding format—the equivalent bit-width is simply the ratio between the high-dimensional lifted space and the target weight space. By marginally adjusting the size of this lifted dimension, LiftQuant can modulate the compression rate with fine-grained precision, naturally yielding contin-

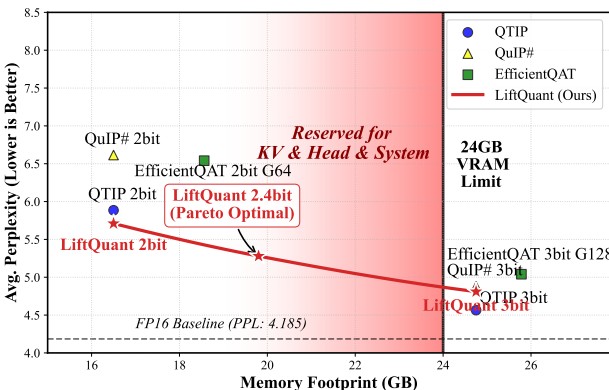

*Figure 1.* Pareto-Optimal Deployment on a 24GB GPU. Perplexity (WikiText-2 and C4) vs. Memory Footprint for Llama-3-70B. While advanced integer-based methods like QTIP and EfficientQAT leave memory wasted or exceed the limit, LiftQuant enables a 2.4-bit model that fully utilizes the available VRAM, significantly outperforming 2-bit baselines. Note that the reserved memory buffer (red zone) is dynamic, varying with deployment scenarios (e.g., KV cache length and precision, batch size, lm.head precision). LiftQuant allows for flexible bit-width tuning to precisely match the remaining available memory.

uous, fractional bit-widths without altering the underlying quantization operator.

This paradigm shift offers a "best-of-both-worlds" solution. The projection generates a structured, non-uniform quantization that rivals the expressive power of vector codebook, yet the decoding process relies solely on a low-complexity linear transformation and a 1-bit uniform quantizer. Our extensive experiments demonstrate that LiftQuant not only matches state-of-the-art integer quantization methods but, more importantly, dominates the Pareto frontier in practical deployment scenarios. For instance, LiftQuant enables a 70B model to be compressed to 2.4 bits to fit precisely within a 24GB GPU (Figure 1), and similarly allows a 32B model to be deployed at 2.5 bits on a widely available 12GB GPU.

Our main contributions are summarized as follows:

- **Continuous Bit-Width Control for Pareto Optimality**: We propose LiftQuant, the novelty framework to enable continuous bit-width adjustment by decoupling quantization from integer grids. This flexibility allows models to fully utilize available hardware memory, achieving true Pareto-optimal deployment.

- **High-Dimensional Non-Uniformity**: We introduce a "lift-then-project" mechanism that procedurally generates structured, non-uniform codebooks from high-dimensional space. This approach captures the ex-

pressive power of Vector Quantization (VQ), enabling LiftQuant to match or surpass the accuracy of state-of-the-art VQ methods.

- **Unified, Hardware-Friendly Inference Architecture**: We achieve this high accuracy with a decoding path that relies solely on low-complexity linear transformations and Int1 uniform quantizers, which provides a single, unified operator that supports arbitrary precision configurations, simplifying engineering deployment.

## 2. Related Work

Weight-only quantization has emerged as one of the most effective strategies for deploying large language models (LLMs) under strict memory and latency constraints.

Uniform Scalar Quantization (UQ) is the most widely used approach, where a floating-point weight vector $w$ is represented as $w_q \cdot s$, with $w_q$ storing low-bit integer values and $s$ being a floating-point scaling factor. Due to the non-uniform value distribution of LLM weights, recent UQ methods introduce lightweight preprocessing to make weights more amenable to quantization. These include group-wise quantization to preserve important channels (e.g., AWQ (Lin et al., 2024)), low-rank error compensation (e.g., QLoRA (Dettmers et al., 2023), (Liu et al., 2025)), and matrix-based transforms to reshape weight distributions (e.g., QuIP# (Tseng et al., 2024a), Quarot (Ashkboos et al., 2024), SpinQuant (Liu et al., 2024b), FlatQuant(Sun et al., 2024)).

Non-uniform Quantization methods improve performance by creating specialized codebooks. These can be scalar-based, using data-driven levels (e.g., NF4 (Dettmers et al., 2023)) or additive basis vectors (e.g., BCQ (Xu et al., 2018; Park et al., 2025)), but they miss inter-dimensional correlations. Vector Quantization (VQ) addresses this by mapping weight vectors to a learned codebook, exploiting inter-element correlations for superior accuracy in ultra-low-bit regimes (e.g., AQLM (Egiazarian et al., 2024), VPTQ (Liu et al., 2024a), QTIP (Tseng et al., 2024b)). However, VQ's reliance on large, hardware-unfriendly lookup tables imposes significant decoding overhead.

The Inflexibility of Integer Bit-Widths. Despite their diversity, a critical limitation unites these methods: their reliance on rigid, integer-based bit-widths (e.g., 2, 3, 4-bit). This inflexibility prevents models from being optimally fitted to specific hardware memory budgets. While some workarounds exist, they are fundamentally constrained. For instance, UQ methods can coarsely modulate the effective bit-width by varying the group size (e.g., from 128 to 64 in EfficientQAT (Chen et al., 2024)), but this provides only a few discrete "gears" to shift between, not a continuous spectrum. Other approaches achieve specific fractional bit-

widths by using non-power-of-two codebooks (e.g., ternary quantization ~1.58bit (Wang et al., 2025)), but require specialized, non-standard kernels. Most notably, Q-Palette (Lee & Song, 2025) recently proposed a collection of fractional-bit quantizers. However, it achieves fractional bits by assembling a heterogeneous mix of different quantizers (scalar, vector, trellis), which necessitates maintaining a complex library of specialized kernels for each configuration. In contrast, our LiftQuant enables continuous bit-width control through a single, unified, and hardware-friendly architecture. By simply tuning the projection dimension, we achieve arbitrary bit-widths without changing the underlying operator.

# 3. LiftQuant: Continuous Bit Width Control Via Lifted Projection

Current quantization paradigms are trapped in a rigid coupling between representation capacity and integer bit-widths. Whether using scalar grids (UQ) or vector codebooks (VQ), the effective bit-rate is determined by discrete design choices—such as the number of grid points or codebook size—which cannot be smoothly adjusted. This rigidity creates the "deployment gap" discussed in Section 1 and Figure 1, preventing models from optimally utilizing available hardware memory. Furthermore, while VQ offers superior accuracy through non-uniform quantization, its reliance on lookup tables (LUTs) introduces significant latency and engineering complexity, making it difficult to deploy efficiently.

**Key Insight: Decoupling Bit-Width from Coding Format.**
Our key insight is that we can decouple the effective bit-width from the coding format by shifting the quantization process to a higher-dimensional space. Instead of quantizing directly in the target weight space $\mathbb{R}^d$, we propose to represent weights as the projection of a simple, 1-bit uniform lattice from a higher-dimensional "lifted" space $\mathbb{R}^D$.

Crucially, this "lift-then-project" mechanism transforms the bit-width from a discrete architectural constant into a continuous, tunable ratio $D/d$. By simply adjusting the dimension $D$, we can achieve any desired fractional bit-width (e.g., 24/10 = 2.4-bit) without changing the underlying 1-bit quantization operator. As visualized in Figure 2, this linear projection naturally generates a dense, Gaussian-like codebook in the target space, thereby capturing the expressive power of VQ while retaining the hardware efficiency of a simple matrix multiplication.

Based on these principles, LiftQuant operates in three phases. In Section 3.1, we learn the global projection matrix $M$ that defines the fractional bit-width and codebook structure. In Section 3.2, we introduce a lightweight, layer-wise whitening transform $T$ to reshape weights into the i.i.d. Gaussian distribution required by our projec-

tion. Finally, in Section 3.3, we detail the quantization and decoding pipeline, where the fused operation $o = diag(s)W(MTa)$ enables efficient inference.

## 3.1. Projection from Lifted-Space to Subspace

Our approach is grounded in the asymptotic properties of high-dimensional geometry. Specifically, a corollary of the Central Limit Theorem (CLT) states that the linear projection of a high-dimensional hypercubic lattice (i.e., independent Bernoulli variables) onto a lower-dimensional subspace converges to a Gaussian distribution as the dimension increases (Diaconis & Freedman, 1984).

Formally, for a weight vector $w \simeq Mw_q$, where $w \in \mathbb{R}^d$ and $w_q \in \{+1, -1\}^D$, each element $w_i = \sum_{j=1}^{D} M_{ij}y_j$ represents a sum of independent random variables. We call the $M$ Mapping Matrix. Consequently, the resulting codebook naturally forms a dense, Gaussian-like cloud in the target space. This provides a strong theoretical justification for our method: LiftQuant does not merely "learn" to fit the Gaussian weights of LLMs; it structurally generates a Gaussian prior by design.

**Optimization of the $M$.** While the CLT guarantees asymptotic Gaussianity, practical deployment requires a finite and relatively small lifted dimension $D$ to maintain computational efficiency. In this regime, the convergence to a perfect Gaussian is incomplete. To bridge this gap, we explicitly optimize the $M$ to minimize the quantization error on a standard Gaussian distribution. We initialize the rows of $M$ as orthonormal vectors to encourage uncorrelated projections. The matrix is then trained on $\mathcal{W}_\mathcal{N} \sim \mathcal{N}(0, I_d)$:

$$M^* = \arg\min_M \mathbb{E}_{w \sim \mathcal{N}} \left[ \min_{w_q \in \{-1,+1\}^{d_s \cdot b}} \|w - Mw_q\| \right], \tag{1}$$

where the inner minimization represents the nearest-neighbor search (quantization). During training, we approximate the non-differentiable $arg\,min$ operation with a temperature of 10 to enable gradient-based optimization.

**Accelerated Nearest-Neighbor Search.** A key challenge in training $M$ is the exponential complexity of the exact nearest-neighbor search, which scales with $2^D$. To accelerate this, we employ a heuristic search strategy based on the generalized inverse. Specifically, we pad the target vector $w$ with a $(D-d)$-dimensional auxiliary vector $z \in \{+1, -1\}^{D-d}$, and project it back to the lifted space $\mathbb{R}^D$ using the pseudo-inverse of $M$ (derived via SVD). This provides a high-quality initialization for local search, effectively reducing the search space to $2^{D-d}$. This heuristic is also applied during the quantization process (mapping $w$ to $w_q$).

**Dimensionality Constraints and Operating Range** The choice of dimensions $D$ and $d$ involves a trade-off between

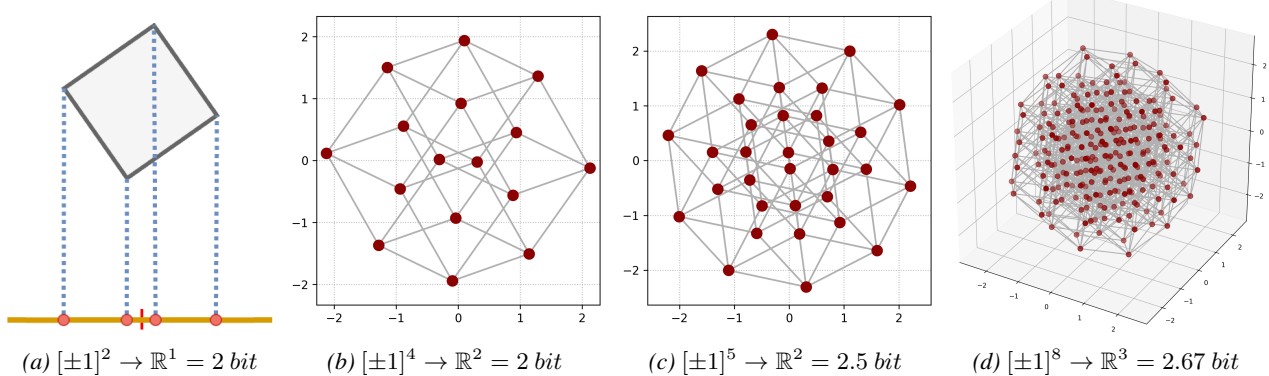

*(a) $[\pm 1]^2 \to \mathbb{R}^1 = 2\ bit$*   *(b) $[\pm 1]^4 \to \mathbb{R}^2 = 2\ bit$*   *(c) $[\pm 1]^5 \to \mathbb{R}^2 = 2.5\ bit$*   *(d) $[\pm 1]^8 \to \mathbb{R}^3 = 2.67\ bit$*

*Figure 2.* Visualization of Codewords Generation in LiftQuant. Our method generates a structured, non-uniform codebook by projecting a simple, uniform lattice from a high-dimensional "lifted" space onto a lower-dimensional target subspace.

Quantized Tensor   **Mapping Matrix**   Dequantized Tensor

| -1 | -1 | 1 | 1 |
| 1 | -1 | 1 | 1 |
| 1. | 1 | -1 | 1 |

$\times$
$\begin{bmatrix} 1.58 & 2.40 \\ 0.33 & -1.54 \\ -2.13 & 0.64 \\ 0.45 & -2.71 \end{bmatrix}$
$\rightarrow$

| -3.59 | -2.93 |
| -0.13 | 1.87 |
| 4.49 | -2.49 |

*Figure 3.* The LiftQuant Dequant Mechanism. A 1-bit quantized tensor in the high-dimensional lifted space is projected via the mapping matrix $M$ to generate the de-quantized weight tensor.

*Table 1.* Comparison of quantization efficiency on a standard Gaussian source and Llama-2-7B perplexity on wikitext2(CTX=2048). Info. bits are derived from $R(D) = 0.5\ log_2(1/MSE)$. Search time (S.T.) denotes the nearest-neighbor search time per 1M parameters. Although LiftQuant's coding efficiency at strictly 2.0 bits is slightly lower than the complex Trellis Coding Quantization used in QTIP, a marginal increase in bit-width (from 32/16 to 30/14) allows LiftQuant to surpass QTIP's performance. Since such minor increments are permissible under most hardware constraints, larger increases yield even greater gains, demonstrating the practical superiority of LiftQuant.

| CODING | BITS | MSE | INFO. | PPL. | S.T. |
|---|---|---|---|---|---|
| LQ-32/20 | 1.60 | 0.146 | 1.39 | 7.71 | 0.3s |
| LQ-16/8 | 2.00 | 0.089 | 1.75 | 6.60 | $\ll$0.1s |
| LQ-32/16 | 2.00 | 0.082 | 1.79 | 6.53 | 4s |
| LQ-30/14 | 2.14 | 0.070 | 1.92 | 6.30 | 4s |
| LQ-24/10 | 2.40 | 0.053 | 2.12 | 6.10 | 1s |
| INT2 | 2.00 | 0.119 | 1.53 | 7.62 | - |
| E8(QUIP#) | 2.00 | 0.089 | 1.75 | 6.60 | - |
| TCQ(QTIP) | 2.00 | 0.073 | 1.89 | 6.28 | - |

representational capacity and search complexity. Generally, higher coding dimensions yield better coding efficiency; for instance, as shown in Table 1, LiftQuant-16/8 is comparable to the E8 lattice, while scaling up to LiftQuant-32/16 surpasses it. However, this comes at the cost of exponentially increased search overhead ($2^{D-d}$).Consequently, under practical search constraints ($D - d <= 20$), LiftQuant cannot strictly match the theoretical efficiency of QTIP at the same bit-width, as QTIP leverages Trellis Codes to handle significantly larger dimensions 64.

Crucially, however, LiftQuant's flexibility offers a superior practical solution: by merely increasing the bit-width from 2-bits to 2.14-bits, its coding efficiency surpasses that of the complex Trellis Codes employed in QTIP. This demonstrates the unique benefit of continuous modulation: we can trade a negligible amount of memory for a simpler, faster, and more flexible decoding architecture, ultimately achieving better Pareto-optimal results.

**Discussion on High-Bit Regimes ($\approx$4-bit)**: As the target bit-width increases (e.g., 4-bit), maintaining a computationally feasible search space ($D - d \le 20$) requires reducing the block dimension $d$ significantly (e.g., $d \le 6$). While technically possible, such small block sizes limit the ability to exploit high-dimensional inter-channel correlations.

This is also a fundamental geometric constraint shared by all VQ methods; for example, AQLM and QuIP# address this by splitting codebooks, which degrades the theoretical coding gain. More importantly, we argue that fine-grained modulation is less critical in this regime. Since recent 4-bit quantization methods already achieve near-lossless performance, the marginal utility of fractional adjustments (e.g., 4.2-bit) is negligible; if needed, coarse-grained schemes (e.g., group-wise quantization) are sufficient. Therefore, LiftQuant strategically focuses on the 2-to-3-bit "deployment gap", where its continuous modulation capability delivers the highest practical value. Even when compared to complex trellis-coded schemes QTIP, LiftQuant maintains a competitive rate-distortion performance (within 0.1 bit gap) while offering superior flexibility for Pareto-optimal deployment.

**Notation Convention.** Since the performance and computational characteristics of LiftQuant are intrinsically tied to the

$D$ and $d$, we adopt a notation of LQ-$D/d$ (e.g., LQ-24/10) throughout the remainder of this paper.

## 3.2. Whitening Transformation

In Section 3.1, we established that our mapping matrix $M$ is optimized to quantize an ideal i.i.d. Gaussian source: $w_\mathcal{N} \simeq M w_q$. However, real-world LLM weights deviate significantly from this assumption. They often exhibit heavy-tailed distributions with outliers (Dettmers et al., 2022) and channel-wise variations in importance due to activation magnitudes (Lin et al., 2024).

To address this mismatch, two primary strategies exist. One is mixed-scheme quantization, which isolates and independently stores salient weights using higher-precision formats (e.g., GPT.int8 (Dettmers et al., 2022), VPTQ (Liu et al., 2024a), QLoRA (Dettmers et al., 2023)). While theoretically efficient in bit-rate, this approach introduces irregular memory access patterns that significantly degrade inference latency. The alternative is distribution reshaping, which applies low-complexity linear transforms to smooth or flatten the weight distribution (e.g., SmoothQuant (Xiao et al., 2022), FlatQuant (Sun et al., 2024)) or to decorrelate channels (e.g., QuIP (Chee et al., 2023), QTIP (Tseng et al., 2024b)). We adopt the latter strategy for its superior hardware efficiency, as linear transforms can be fused or executed in parallel. We term this process "Whitening", borrowing from information theory where source data is transformed to resemble a Gaussian channel input for optimal coding.

**Whitening Transform Design.** To achieve both efficiency and representational power, we parameterize the layer-wise whitening transform $D$ in a decomposed form:

$$T = \operatorname{diag}(s_1)(P_1 \otimes P_2)\operatorname{diag}(s_2) \quad (2)$$

where activation multiplication by $T^{-1}$ scales as $\mathcal{O}(n\sqrt{n})$, significantly lower than the $\mathcal{O}(n^2)$ cost of dense matrix multiplication. Here, $s_{1,2}$ are diagonal scaling matrices, and $P_{1,2}$ are $\sqrt{n} \times \sqrt{n}$ matrices whose Kronecker product provides channel intermixing and whitening capability. This decomposition is extremely parameter-efficient. For a Llama-70B model, storing these transformation parameters in FP16 adds only a negligible overhead of 0.008–0.011 bits per parameter (under 1.6-bit to 3-bit quantization settings).

This design serves three specific functions: 1) Importance-Aware Scaling $s_1$. Inspired by AWQ, $s_1$ redistributes quantization error based on activation magnitudes, down-scaling channels with large activations to reduce their relative error. 2) Decorrelation and Mixing transformation $P_{1,2}$. We initialize them as orthogonal matrices (e.g., Hadamard) to preserve energy and diffuse outliers across dimensions, similar to QuIP#. 3) Isotropy Refinement $s_2$. The final scal-

ing $s_2$ further normalizes per-channel variance, ensuring stronger isotropy with respect to the LiftQuant lattice. Crucially, by constraining $s_2$ to be block-wise constant, it can be fused with the projection matrix $M$ during inference (see Section 3.3).

The optimization objective for $T$ is to minimize the reconstruction error under the reversible transformation:

$$\arg \min_{s_{1,2}, P_{1,2}} \mathcal{L}\|W a^T - \operatorname{Quant}_{\text{ste}}(WT)T^{-1}a^T\|^2 \quad (3)$$

where $\operatorname{Quant}_{\text{ste}}$ denotes a standard uniform quantizer with Straight-Through Estimator (STE). We employ this standard quantizer as a proxy because distributions amenable to uniform quantization—specifically, those that are decorrelated and outlier-free—are inherently compatible with our Gaussian-optimized LiftQuant projection. This proxy approach not only ensures high computational efficiency during training but also enhances robustness by avoiding the optimization instability often associated with complex non-uniform quantizers. This decomposed transform effectively reshapes arbitrary weight distributions into well-approximated i.i.d. Gaussians, enabling the seamless application of our LiftQuant projection.

## 3.3. Quantization in Lifted-Space and Intra-Block Correction

Having obtained the projection matrix $M$ and the whitening transform $T$, we integrate them into a unified quantization pipeline. First, the layerweights $W$ are whitened and standardized to match the distribution expected by the LiftQuant lattice. These pre-processed weights are then quantized by finding their nearest neighbors in the lifted space, yielding a binary matrix $W_q \in \{+1, -1\}^{OC \times (\lceil \frac{IC}{d} \rceil \cdot D)}$.

Crucially, for efficient inference, the projection matrix $M$ and the inverse whitening transform $T^{-1}$ can be mathematically fused into a single linear operator. The layer output is thus computed as:

$$o = \operatorname{diag}(s) \cdot W_q(MT^{-1}a^T) = \operatorname{diag}(s) \cdot W_q(T^*a^T) \quad (4)$$

where $s$ is the quantization scale factor, and $T^* = MT^{-1}$ represents the fused decoding matrix. This formulation reveals that the entire decoding process is reduced to a low-complexity linear projection followed by a Int1-Float matrix multiplication, ensuring high throughput and hardware-friendly deployment.

A key advantage of this fused formulation is that it renders the entire quantization-decoding path fully differentiable. This allows us to further refine the model performance through block-wise fine-tuning. Specifically, we treat the binary weights $W_q$ (via Straight-Through Estimator) and the fused matrix $T^*$ as trainable parameters. We

optimize them to minimize the reconstruction error of the layer output using a small calibration dataset:

$$\min_{\boldsymbol{W}_q, \boldsymbol{T}^*} \mathbb{E}_{\boldsymbol{a} \sim \mathcal{T}_{\text{calib}}} ||\boldsymbol{W}_{\text{fp}} \boldsymbol{a}^T - \text{diag}(\boldsymbol{s}) \boldsymbol{W}_q \boldsymbol{T}^* \boldsymbol{a}^T||^2 \quad (5)$$

This local adaptation step is critical for compensating for any residual quantization errors and aligning the components for optimal end-to-end performance.

## 4. Experiments

**Setting**. We evaluate LiftQuant on a comprehensive suite of LLMs, including the Llama-2/3 (Touvron et al., 2023), and Qwen-2.5/3 (Yang et al., 2025) families. We report perplexity (PPL) on WikiText-2(Merity et al., 2016) and C4(Raffel et al., 2020) validation sets, zero-shot accuracy on five common-sense reasoning benchmarks (ARC-c, ARC-e(Clark et al., 2018), HellaSwag(Zellers et al., 2019), PIQA(Bisk et al., 2020), WinoGrande(Sakaguchi et al., 2021)), and MMLU(Hendrycks et al., 2020). We compare against state-of-the-art methods, including leading UQ approaches (GPTQ, Quarot, EfficientQAT) and advanced VQ frameworks (QuIP#, AQLM, VPTQ, QTIP). For calibration, we sample 4,096 segments with 2,048 sequence length, from the RedPajama dataset. More calibration details are provided in Section A.

**End-to-End Fine-Tuning**. Following the standard protocol adopted by recent top-performing methods (e.g., AQLM, QuIP#, EfficientQAT), we perform a lightweight end-to-end fine-tuning step after quantization. This process optimizes the continuous quantization parameters (e.g., scales, transformation matrices) by minimizing the cross-entropy loss on a small calibration set, ensuring fair comparison. Further details are provided in Section B. And the sensitivity of our method to the calibration data is discussed in Section C.

### 4.1. Main Results

As shown in Tables 2 and 3, LiftQuant demonstrates a substantial performance advantage over all UQ baselines. Even compared to methods that employ fine-grained grouping (e.g., EfficientQAT-g64), LiftQuant consistently achieves lower perplexity and higher accuracy. This validates that our "lift-then-project" mechanism successfully overcomes the inherent limitations of the uniform grid, capturing the non-uniform weight distribution far more effectively than scalar approaches.

Against state-of-the-art VQ methods under standard integer bit-widths, LiftQuant delivers highly competitive performance. On most models, LiftQuant matches the accuracy of leading VQ frameworks like QuIP# and AQLM. While QTIP achieves slightly better theoretical coding efficiency due to its trellis-coded structure, LiftQuant remains within a narrow margin. But on the Llama-3-70B model, LiftQuant achieves a 2-bit perplexity of 5.31, slightly outperforming QTIP. We attribute this to LiftQuant's fully differentiable architecture, compensating for the theoretical gap in quantization error.

Most critically, LiftQuant unlocks a new performance tier through **fractional quantization**. As shown in the results, simply increasing the bit-width to 2.4 bits allows LiftQuant to significantly outperform all 2-bit baselines (both UQ and VQ) by a large margin. For example, on Llama-3-70B, the 2.4-bit LiftQuant model achieves a perplexity of 5.86, far surpassing the best 2-bit result of 5.31. This demonstrates that in practical scenarios where memory budgets allow for slightly more than 2 bits (e.g., 24GB VRAM), LiftQuant's ability to utilize that extra capacity provides a decisive advantage that no integer-constrained method can match.

### 4.2. Fractional Bit Widths and Pareto Optimality

To demonstrate the full potential of LiftQuant, we conducted an extensive evaluation across the Llama-2, Qwen-2.5, and Qwen-3 families, covering model sizes from 3B to 70B. We measured zero-shot accuracy and MMLU performance across a continuous spectrum of bit-widths, plotting these against memory footprint to construct the Pareto frontier.

To contextualize our results, we introduce an "Ideal 4-Bit" reference point. Recent foundational models, such as GPT-oss-120B and Kimi-K2, have adopted the MXFP4 format (effective bit-width 4.25 bits) as their native representation, achieving performance indistinguishable from FP16. We thus assume a hypothetical 4-bit quantization that matches FP16 accuracy to serve as the upper bound for compression efficiency.

Figure 4 illustrates the performance-memory trade-off curves. Our findings reveal three critical insights:

1. Alignment with the Ideal Frontier: We observe that the Pareto frontier formed by LiftQuant models across various fractional bit-widths closely aligns with the curve formed by the "Ideal 4-Bit" reference points. This suggests that a LiftQuant-compressed model approximates the performance profile of a native model of that effective size. In this sense, LiftQuant can be viewed as a flexible "parameter scaler", offering a practical means to instantiate models of intermediate effective sizes to match hardware constraints, potentially reducing the need to pre-train dense models for every specific memory budget.

2. Effective Deployment Strategies: This flexibility facilitates better hardware matching. As highlighted in Figure 1, we demonstrate the feasibility of deploying a 70B model at 2.4 bits (24/10) on a 24GB GPU, and a 32B model at 2.5 bits (25/10) on a 12GB GPU. In our experiments, these configurations maximized VRAM utilization and delivered

*Table 2.* Perplexity (↓) on Wikitext2 and C4, context length 2048 for Llama-2 and 8192 for Llama-3.

| Method | Type | Bits | 2-7 | | 2-13 | | 2-70 | | 3-8 | | 3-70 | |
|---|---|---|---|---|---|---|---|---|---|---|---|---|
| | | | W2 | C4 | W2 | C4 | W2 | C4 | W2 | C4 | W2 | C4 |
| FP16 | - | - | 5.47 | 6.97 | 4.88 | 6.47 | 3.32 | 5.52 | 5.54 | 7.10 | 2.59 | 5.78 |
| GPTQ-g128 | UQ | 2.13 | 50.75 | 36.76 | 43.84 | 23.07 | NaN | NaN | - | - | - | - |
| Quarot | UQ | 2.00 | 22.07 | - | 10.41 | - | 5.60 | - | - | - | - | - |
| OmniQ-g64 | UQ | 2.25 | 9.62 | 12.72 | 7.56 | 10.05 | 6.11 | 7.68 | - | - | - | - |
| EQAT-g64 | UQ | 2.25 | 6.86 | 8.50 | 5.96 | 7.59 | 4.52 | 6.38 | 8.31 | 10.09 | 5.36 | 7.43 |
| AQLM | VQ | 1.97-2.07 | 6.61 | 8.28 | 5.72 | 7.44 | 4.19 | 6.13 | - | - | - | - |
| VPTQ | VQ | 2.02-2.08 | 6.57 | 8.27 | 5.69 | 7.41 | 4.17 | 6.13 | - | - | 5.55 | 7.30 |
| QuIP# | VQ | 2.00 | 6.66 | 8.35 | 5.74 | 7.45 | 4.16 | 6.12 | 7.84 | 9.06 | 5.77 | 7.46 |
| QTIP | VQ | 2.00 | 6.28 | 7.94 | 5.45 | 7.16 | 3.94 | 5.93 | 7.33 | 8.62 | 4.97 | 6.80 |
| LiftQuant | LQ-32/16 | 2.01-2.02 | 6.52 | 8.21 | 5.61 | 7.34 | 4.08 | 6.06 | 7.65 | 8.95 | 4.69 | 6.73 |
| LiftQuant | LQ-24/10 | 2.41-2.42 | 6.10 | 7.70 | 5.30 | 6.98 | 3.78 | 5.83 | 6.94 | 8.31 | 4.10 | 6.47 |
| GPTQ | UQ | 3.00 | 8.37 | 9.81 | 6.44 | 8.02 | 4.82 | 6.57 | - | - | - | - |
| GPTQ-g128 | UQ | 3.13 | 6.29 | 7.89 | 5.42 | 7.00 | 3.85 | 5.85 | 8.81 | 9.35 | 4.82 | 7.37 |
| Quarot | UQ | 3.00 | 6.09 | - | 5.37 | - | 3.72 | - | - | - | - | - |
| EQAT-g128 | UQ | 3.13 | 5.81 | 7.34 | 5.12 | 6.73 | 3.61 | 5.71 | 6.35 | 7.79 | 3.78 | 6.30 |
| VPTQ# | VQ | 3.01-3.03 | 5.82 | 7.33 | 5.12 | 6.70 | 3.55 | 5.67 | - | - | - | - |
| QuIP# | VQ | 3.00 | 5.79 | 7.32 | 5.10 | 6.72 | 3.56 | 5.67 | 6.27 | 7.71 | 3.59 | 6.18 |
| LiftQuant | LQ-24/8 | 3.01-3.02 | 5.75 | 7.31 | 5.09 | 6.71 | 3.35 | 5.67 | 6.22 | 7.70 | 3.45 | 6.18 |

*Table 3.* Llama accuracy(↑) on 2-bit quantization(LQ-32/16).

| Method | bits | ArcC | ArcE | Hella | PiQA | Wino | Avg. |
|---|---|---|---|---|---|---|---|
| 2-7 | - | 43.52 | 76.26 | 57.16 | 78.07 | 69.22 | 64.85 |
| EQAT-g64 | 2.25 | 36.86 | 70.96 | 51.58 | 75.30 | 65.98 | 60.14 |
| QuIP# | 2.00 | 37.88 | 71.84 | 50.84 | 74.16 | 65.67 | 60.61 |
| QTIP | 2.00 | 39.76 | 73.32 | 53.68 | 76.28 | 67.25 | 62.06 |
| LiftQuant | 2.02 | 36.77 | 70.66 | 53.05 | 76.55 | 68.27 | 61.06 |
| LiftQuant | 2.42 | 39.42 | 72.90 | 55.10 | 76.99 | 67.72 | 62.42 |
| 2-13 | - | 48.29 | 79.42 | 60.07 | 79.05 | 72.22 | 67.81 |
| EQAT-g64 | 2.25 | 41.89 | 74.83 | 55.27 | 77.04 | 68.36 | 63.48 |
| QuIP# | 2.00 | 42.92 | 75.72 | 56.53 | 77.97 | 69.06 | 64.44 |
| QTIP | 2.00 | 43.94 | 76.60 | 58.13 | 77.69 | 70.40 | 65.35 |
| LiftQuant | 2.02 | 43.69 | 76.30 | 57.09 | 77.91 | 70.01 | 65.00 |
| LiftQuant | 2.42 | 44.80 | 77.48 | 58.64 | 77.48 | 71.27 | 65.93 |
| 2-70 | - | 54.44 | 82.70 | 64.77 | 82.15 | 77.98 | 72.41 |
| EQAT-g64 | 2.26 | 50.77 | 80.13 | 61.78 | 80.14 | 74.59 | 69.48 |
| QuIP# | 2.00 | 52.65 | 81.90 | 62.86 | 81.39 | 75.77 | 70.91 |
| QTIP | 2.00 | 53.05 | 81.12 | 63.11 | 82.16 | 76.19 | 71.12 |
| LiftQuant | 2.01 | 53.58 | 81.36 | 63.08 | 81.28 | 75.69 | 71.00 |
| LiftQuant | 2.41 | 51.62 | 81.52 | 64.31 | 82.32 | 77.19 | 71.39 |
| 3-8 | - | 50.43 | 80.09 | 60.17 | 79.60 | 72.61 | 68.58 |
| EQAT-g64 | 2.25 | 37.03 | 71.17 | 51.86 | 76.03 | 67.72 | 60.76 |
| VPTQ | 2.07 | 36.91 | 71.03 | 52.12 | 75.12 | 65.92 | 60.22 |
| LiftQuant | 2.02 | 40.87 | 74.33 | 53.87 | 76.55 | 68.03 | 62.73 |
| 3-70 | - | 60.41 | 86.99 | 66.36 | 82.37 | 80.51 | 75.33 |
| EQAT-g64 | 2.25 | 49.06 | 77.40 | 61.60 | 77.37 | 74.03 | 67.89 |
| VPTQ | 2.02 | 52.65 | 81.86 | 61.71 | 80.36 | 77.90 | 70.90 |
| LiftQuant | 2.01 | 56.14 | 84.30 | 62.31 | 81.72 | 78.53 | 72.60 |

*Table 4.* E2E decoding for Llama-2-70B on GTX4090D-48G. Context Length = 512, batch size = 1.

| LIFTQUANT | LQ-32/16 | LQ-24/10 | LQ-24/8 |
|---|---|---|---|
| | 31.3 TK/S | 25.7 TK/S | 20.8 TK/S |
| QTIP | 2BIT | - | 3BIT |
| | 24.5 TK/S | - | 17.6 TK/S |
| AWQ | 2BIT | - | - |
| | 36.1 TK/S | - | - |

performance that consistently exceeded that of smaller models quantized to standard integer bit-widths.

3. The "2x Scaling + Quantization" Heuristic: Our data indicates that performance tends to deviate noticeably from the Pareto line when the bit-width drops below 2 bits. However, the 2-to-4-bit range generally adheres to the optimal frontier. This observation points towards a potential deployment heuristic: by scaling native model sizes in steps of roughly 2x (e.g., 7B → 14B → 32B) and using LiftQuant to bridge the 2-to-4-bit gap, it may be possible to construct a dense and near-optimal deployment continuum.

### 4.3. Inference Efficiency

To validate the practical efficiency of LiftQuant, we evaluated the decoding throughput of the Llama-2-70B model on an NVIDIA RTX 4090D (48GB) GPU (Table 4). A significant advantage of LiftQuant is its architectural simplicity.

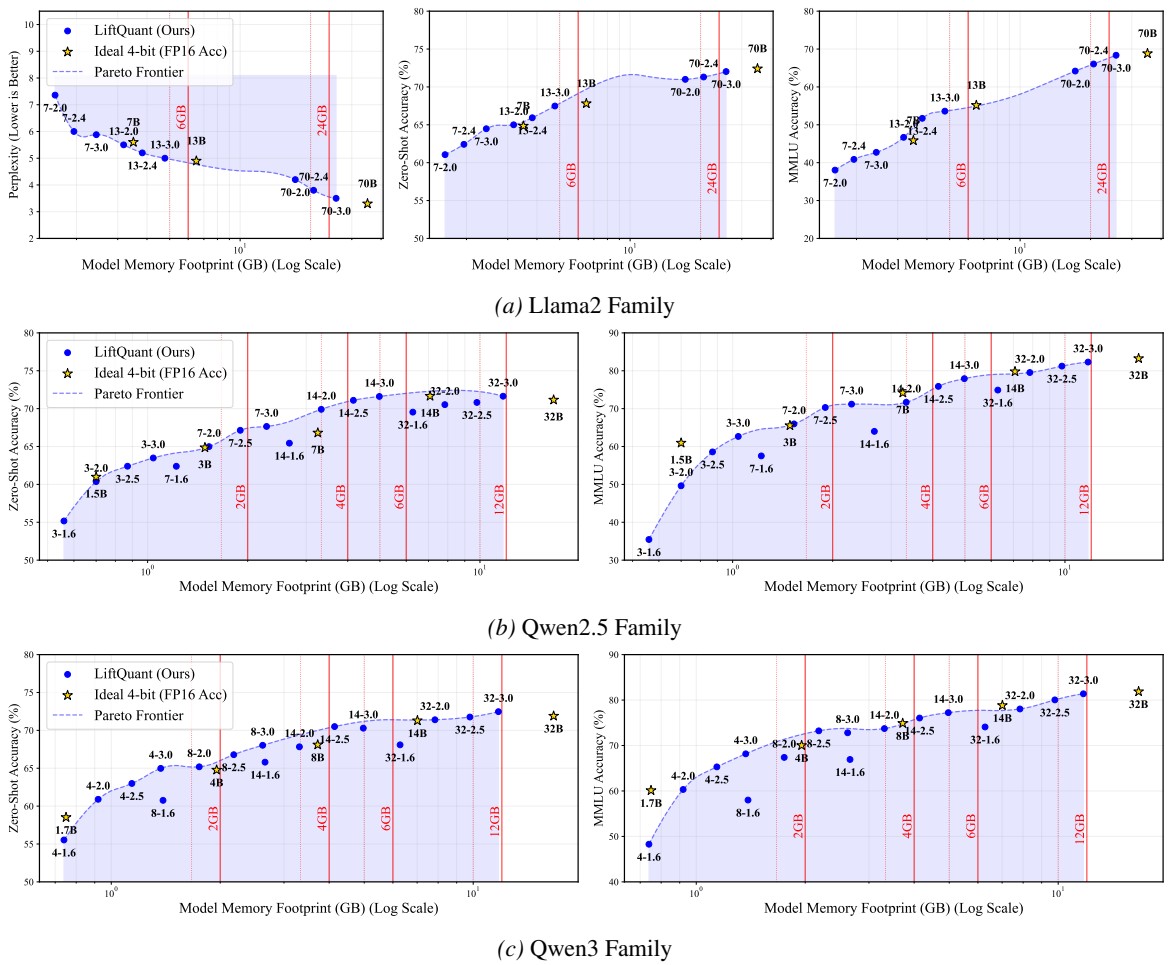

*(a)* Llama2 Family

*(b)* Qwen2.5 Family

*(c)* Qwen3 Family

*Figure 4.* Performance (PPL, Zero-shot, MMLU) vs. Memory Footprint across multiple model families. LiftQuant's fractional bit widths fill the gaps between integer steps, creating a dense frontier that enables customized, optimal quantization for arbitrary memory constraint.

We implemented the linear operation $\boldsymbol{o} = diag(\boldsymbol{s})\boldsymbol{W}(\boldsymbol{T}^*\boldsymbol{a})$ efficiently using torch.compile and BitBLAS (Wang et al., 2024) for UINT1-FP16 GEMV operations.

To the best of our knowledge, while QTIP employ specialized kernels heavily optimized for batch size 1, it often lack native support for other settings (e.g., batch size 8). So it requires substantial engineering effort to develop dedicated kernels for each scenario. In contrast, LiftQuant achieves ideal acceleration across diverse settings using only standard open-source libraries.

### 4.4. Ablation Study

To validate the contribution of each component in the LiftQuant framework, we conducted a progressive ablation study on the Llama-2-7B model. We evaluated the impact of the whitening transform parameters ($\boldsymbol{s}_1$, $\boldsymbol{P}_1$, $\boldsymbol{P}_2$, $\boldsymbol{s}_2$ in $\boldsymbol{T}$) and the projection matrix . The results are reported as perplexity on WikiText-2 in Table 5.

*Table 5.* Ablation Study on Llama-2-7b

| COMPONENT | $\boldsymbol{P}_1 + \boldsymbol{P}_2$ | $+\boldsymbol{s}_1$ | $+\boldsymbol{s}_2$ | $+\boldsymbol{M}$ | + F.T. |
|---|---|---|---|---|---|
| WIKI2 PPL($\downarrow$) | 8.76 | 8.28 | 7.77 | 6.79 | 6.53 |

When $\boldsymbol{M}$ is not used, we default to a standard 2-bit symmetric uniform quantizer. Notably, we found that the matrices $\boldsymbol{P}_1$ and $\boldsymbol{P}_2$ (initialized as random orthogonal matrices) are critical for stability; omitting them leads directly to training collapse. Thus, our ablation starts with $\boldsymbol{P}_1$ and $\boldsymbol{P}_2$ enabled. As shown in Table 3, adding the scaling factors $\boldsymbol{s}_1$ and $\boldsymbol{s}_2$ progressively improves performance. Introducing the LiftQuant projection matrix $\boldsymbol{M}$ yields a significant accuracy boost, confirming the benefit of high-dimensional quantization. Finally, end-to-end fine-tuning further refines the model, delivering the best overall results.

# 5. Conclusion

We introduced LiftQuant, a framework that breaks the rigidity of integer quantization by enabling continuous bit-width control. Through a "lift-then-project" mechanism, LiftQuant decouples quantization rate from coding format, allowing for true Pareto-optimal deployment. Our experiments confirm that LiftQuant matches state-of-the-art vector quantization methods accuracy, while uniquely enabling fractional configurations that maximize performance on constrained hardware. LiftQuant establishes a practical, hardware-aware paradigm for customized LLM compression.

# Impact Statement

This paper presents work whose goal is to advance the field of Machine Learning. There are many potential societal consequences of our work, none which we feel must be specifically highlighted here.

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

# A. Training Details for Intra-Block correction

This appendix details the training procedure for the block-wise correction phase described in Section 3.3. The goal of this phase is to correct for quantization errors by jointly optimizing the low-bit weights $W_q$ and the transformation matrix $T^*$.

Two primary strategies exist for post-quantization correction. The first, based on the Hessian matrix, involves adaptively rounding weight vectors (Frantar et al., 2022; Tseng et al., 2024a). However, this class of methods is impractical for our framework due to the prohibitive computational cost of the nearest-neighbor search required to determine the set of valid rounding candidates for each vector in our lattice.

Consequently, we adopt a more practical and effective approach: direct fine-tuning using gradient descent (Egiazarian et al., 2024; Chen et al., 2024). This method, proven viable in prior work, allows us to optimize both $T^*$ and $W_q$ simultaneously. Since $W_q$ consists of discrete values, we employ the Straight-Through Estimator (STE) to approximate gradients during backpropagation.

For the correction process, we constructed a calibration dataset by randomly selecting 4,096 samples from the RedPajama dataset, with each sample having a sequence length of 2048 tokens. From this set, 128 samples were held out as a validation set. We used the Adam optimizer to minimize the Mean Squared Error (MSE) loss between the outputs of the quantized layer and the original full-precision layer. The learning rate for the transformation parameters $T^*$ was set to $1 \times 10^{-3}$ across all models. For the $W_q$, we used a learning rate of $2 \times 10^{-5}$ for models between 3B and 14B parameters, and a reduced rate of $1 \times 10^{-5}$ for the 70B model. The entire training process was conducted for 2 epochs.

# B. Training Details for end to end fine-tune

To further enhance model performance and globally align the quantization parameters, we perform an optional end-to-end fine-tuning step. The effectiveness of this approach for adjusting quantization parameters has been validated by several prior works (Tseng et al., 2024a; Egiazarian et al., 2024; Chen et al., 2024; Liu et al., 2024a).

This fine-tuning process optimizes the continuous parameters of our framework—specifically, the scaling parameters and the components of the transformation matrix $D$—across all layers simultaneously. Unlike the layer-wise correction phase, this step minimizes the standard language modeling loss (i.e., Cross-Entropy) over the entire model.

For training, we used a dataset of 4,096 samples from RedPajama, each with a sequence length of 4096. We employed the Adam optimizer and trained for a single epoch. A differential learning rate scheme was applied: the learning rate for the quantization scaling parameters was set to $1 \times 10^{-5}$, while the transformation parameters used a higher rate of $3 \times 10^{-4}$. A significant advantage of this approach is its remarkable memory efficiency. Since the fine-tuning is performed on the already quantized model, the weights remain in their low-bit format throughout the process. This dramatically reduces the memory footprint, enabling us to fine-tune the entire 70B model on a single 80GB A100 GPU—a task that is infeasible for its full-precision counterpart.

# C. Sensitivity to Calibration Data

To investigate the sensitivity of our fine-tuning process to the choice of calibration data, we conducted a comprehensive ablation study. We varied the calibration dataset's size, domain, and sequence length, and evaluated the impact on Llama-2-7B. For these experiments, we used a 10x20 $M$-matrix and only performed the intra-block correction (without end-to-end fine-tuning) to isolate the specific effect of the calibration data. The results are presented in Table 6.

*Table 6.* Ablation study on the calibration data for 2-bit Llama-2-7B. The default configuration used in our main experiments is highlighted in bold.

| Calibration Set | Config. (Samples × SeqLen) | WikiText-2 PPL (↓) | C4 PPL (↓) | Avg. 0-shot Acc. (↑) |
|---|---|---|---|---|
| RedPajama (Small) | 512 × 2048 ( 1M tokens) | 7.08 | 8.66 | 60.03 |
| RedPajama (Medium) | 1024 × 2048 ( 2M tokens) | 7.00 | 8.59 | 60.55 |
| RedPajama (Large) | 2048 × 2048 ( 4M tokens) | 6.96 | 8.53 | 60.68 |
| **RedPajama (Default)** | **4096 × 2048 ( 8M tokens)** | **6.97** | **8.53** | **60.70** |
| RedPajama (Short Seq) | 4096 × 512 ( 2M tokens) | 6.98 | 8.53 | 60.67 |
| WikiText-2 (In-Domain) | 2048 × 2048 ( 4M tokens) | **6.72** | 8.65 | 60.24 |

Our findings from this study provide two key insights:

**Robustness to Data Size and Sequence Length.** The results indicate that while performance improves as the calibration data size increases from 1M to 8M tokens, there are clear diminishing returns beyond approximately 4M tokens. Similarly, reducing the sequence length from 2048 to 512 while keeping the total token count constant has a minimal impact on the final performance. **Our choice of 8M tokens (4096 samples $\times$ 2048 sequence length) for the main experiments was made to ensure a fair comparison with other methods, such as AQLM and EfficientQAT.**

**Impact of Domain Shift.** As expected, calibrating on a domain-matched dataset (WikiText-2) yields the best perplexity on that specific in-domain benchmark (6.72 PPL), as shown in Table 6. This specialization, however, comes at the cost of slightly degraded performance on out-of-domain benchmarks like the C4 dataset and zero-shot tasks. Using a large, general-purpose corpus like RedPajama provides a more balanced and robust performance across all evaluation metrics.

## D. Decoding Overhead

*Table 7.* Asymptotic complexity and storage analysis per layer of size $N \times M$ at 2bit quantization.

| Method | Main GEMM FLOPs | Additional FLOPs | Weight Storage | Additional Storage |
|---|---|---|---|---|
| FP16 | $2NM$ | - | $16NM$ | - |
| **LiftUQ** (decoding) ($\boldsymbol{T^*A}$ first, batch=1) | $2NM$ | $O(d_sN + d_s^2N)$ | $bNM$ | $O(d_s^2N)$ |
| **LiftUQ** (prefill) ($\boldsymbol{W_qT^*}$ first, batch=$k$) | $2kNM$ | $O((d_sN + d_s^2N)k)$ | $bNM$ | $O(d_s^2N)$ |

Note: $k$ is batch size, $b$ is bitwidth, $d_s$ is subspace dimension.

## E. Comparison on 1.58-bit Baseline.

*Table 8.* Comparison on 1.58-bit Baseline.

| Method | Type | Bits | 2-7 | | | 2-13 | | | 3-8 | | |
|---|---|---|---|---|---|---|---|---|---|---|---|
| | | | W2↓ | C4↓ | Avg.Acc↑ | W2↓ | C4↓ | Avg.Acc↑ | W2↓ | C4↓ | Avg.Acc↑ |
| FP16 | - | - | 5.47 | 6.97 | 64.85 | 4.88 | 6.47 | 67.81 | 6.14 | 8.88 | 68.58 |
| PTQ1.61 | UQ | 1.61 | 12.70 | 17.73 | 44.14 | 9.74 | 13.64 | 49.21 | 22.90 | 33.82 | 43.99 |
| LiftUQ | LQ-16/10 | **1.62** | **7.71** | **9.55** | **56.19** | **6.47** | **8.27** | **60.54** | **11.43** | **15.13** | **56.66** |

## F. Other Quantization Result.

*Table 9.* Llama-2 and Llama-3 accuracy(↑) on 3-bit quantization.

| Model | Method | type | bits | ArcC | ArcE | HellaSwag | PiQA | WinoGrande | Avg.Acc |
|-------|--------|------|------|------|------|-----------|------|------------|---------|
| 2-7 | FP16 | - | - | 43.52 | 76.26 | 57.16 | 78.07 | 69.22 | 64.85 |
| | QuIP# | VQ | 3.00 | 41.89 | 74.62 | 55.85 | 77.04 | 68.19 | 63.52 |
| | VPTQ | VQ | 3.02 | 39.3 | 69.1 | 54.9 | 77.3 | 68.0 | 61.70 |
| | LiftUQ | UQ | 3.02 | 41.02 | 75.07 | 56.57 | 77.89 | 67.97 | 63.71 |
| 2-13 | FP16 | - | - | 48.29 | 79.42 | 60.07 | 79.05 | 72.22 | 67.81 |
| | QuIP# | VQ | 3.00 | 44.62 | 77.90 | 58.26 | 78.07 | 72.45 | 66.26 |
| | VPTQ | VQ | 3.03 | 46.50 | 78.83 | 58.50 | 78.18 | 69.85 | 66.37 |
| | LiftUQ | UQ | 3.02 | 46.25 | 77.99 | 59.16 | 78.84 | 71.11 | 66.67 |
| 2-70 | FP16 | - | - | 54.44 | 82.70 | 64.77 | 82.15 | 77.98 | 72.41 |
| | QuIP# | VQ | 3.00 | 55.89 | 82.11 | 64.22 | 82.21 | 76.24 | 72.13 |
| | LiftUQ | UQ | 3.02 | 54.61 | 82.58 | 63.98 | 81.50 | 77.11 | 71.96 |
| 3-8 | FP16 | - | - | 50.43 | 80.09 | 60.17 | 79.60 | 72.61 | 68.58 |
| | VPTQ | VQ | 3.03 | 44.80 | 78.45 | 57.85 | 78.78 | 71.74 | 66.32 |
| | LiftUQ | UQ | 3.02 | 46.59 | 78.83 | 58.42 | 78.73 | 73.95 | 67.30 |
| 3-70 | FP16 | - | - | 60.41 | 86.99 | 66.36 | 82.37 | 80.51 | 75.33 |
| | AWQ-g128 | UQ | 3.13 | 58.36 | 84.51 | 64.26 | 82.26 | 78.85 | 73.65 |
| | EPTQ-g128 | UQ | 3.13 | 55.12 | 83.12 | 65.53 | 80.52 | 77.82 | 72.42 |
| | LiftUQ | UQ | 3.02 | 58.87 | 85.86 | 65.32 | 82.43 | 78.77 | 74.25 |

