# OpenReview forum: "LiftQuant: Continuous Bit-Width LLM via Dimensional Lifting and Projection"
_ICML.cc/2026/Conference — ICML 2026 spotlight_

### Official Review · Reviewer_nSVV · 2026-02-25

**Soundness:** 3
**Presentation:** 3
**Significance:** 3
**Originality:** 3
**Overall Recommendation:** 4
**Confidence:** 4

**Summary:**

LiftQuant introduces a novel quantization framework that replaces rigid integer bit-widths with a continuous fractional-bit design space via a “lift-then-project” mechanism.

By projecting weights from a high-dimensional 1-bit lattice and controlling the dimension ratio, it enables arbitrary bit-widths (e.g., 2.4-bit), generating non-uniform Gaussian-like codebooks that better capture LLM weight distributions while maintaining a hardware-friendly decoding path.

Experimental results on Llama and Qwen model families show that fractional-bit configurations significantly outperform conventional 2-bit baselines in perplexity and zero-shot accuracy, effectively bridging the deployment gap between memory constraints and model performance. The framework also demonstrates competitive efficiency through a unified linear inference pipeline and standard GEMV operations, suggesting practical viability for real-world deployment under resource limits.

**Compliance With Llm Reviewing Policy:**

Affirmed.

**Final Justification:**

As mentioned in final rebuttal, although the author has made some explanations, there are still parts that have not been resolved, thus I believe my current score is appropriate.

**Key Questions For Authors:**

None

**Strengths And Weaknesses:**

## **Strength**
1. **Continuous Fractional-Bit Quantization**: Introduces a novel design space that decouples bit-width from discrete integer choices, enabling fine-grained control over memory–accuracy trade-offs and improved hardware utilization.

2. **Practical VRAM Utilization**: Experimental results show effective utilization of available memory (e.g., 24GB GPUs with 2.4-bit configurations), outperforming smaller integer-quantized models in comparable resource settings.

## **Weakness**

1. **Computational and Search Complexity**: The paper itself notes that when reducing bit-widths to extreme fractional values, it is necessary to operate in a high-dimensional projection space of dimension D/d, which incurs exponential search overhead. To make training feasible, the authors employ heuristic methods (such as auxiliary vectors and pseudo-inverse initialization) to accelerate nearest-neighbor search. However, this still introduces computational overhead risks in theory, especially in very low-bit regimes (where the constraint $D−d≤20$). This implies:
   - Practical deployment requires careful design of search algorithms and training mechanisms, making it less straightforward than standard integer quantization.

2. **Dependence on Calibration Data**
LiftQuant still requires calibration data and end-to-end fine-tuning to optimize continuous quantization parameters. In scenarios where calibration data is insufficient or distribution shifts occur, performance may degrade. When the model structure or weight distribution differs significantly (e.g., heavy-tailed outliers), the pre-whitening transformation may become less stable.
(This is also a challenge in VQ and Q-Palette-style methods, but LiftQuant appears more sensitive to data quality.)

3. **Deployment Practicality vs Standard Hardware**
Although fractional bit-widths theoretically bridge the deployment gap, mainstream hardware and software stacks (e.g., existing operator libraries and low-bit computation support) do not natively support non-integer bit operations. As a result, real-world deployment still requires engineering compromises.
For example, implementing true 2.4-bit GEMM operations would still need to be mapped to standard integer GPU instruction paths at runtime, which is more complex than integer-only pipelines.

---

> ### Author Rebuttal · Authors · 2026-03-31
>
> **Dear Reviewer nSVV,**
>
> We sincerely thank you for your positive evaluation and constructive feedback. We are glad that you recognize the practical value of continuous bit-width quantization.
>
>
> ------
>
> **W1: Computational and search complexity**
>
> We would like to clarify that the search and optimization overhead is entirely confined to the **offline quantization stage**. The mapping matrix $M$ is computed once and reused, and in practice we explicitly constrain the search space (D − d ≤ 20), making it tractable. Our pseudoinverse-based heuristic is also highly reliable. Similar to other ultra-low-bit methods (e.g., EfficientQAT, QTIP), a multi-stage pipeline is necessary to maintain accuracy in the 2–3 bit regime. Importantly, no search is involved during inference.
>
> --------
>
> **W2: Dependence on calibration data**
>
> We agree that extreme data scarcity may affect stability. In such cases, we observe that the whitening components (P1, P2) are the most sensitive, and we recommend fixing them as Hadamard matrices. Under standard calibration settings (~10M tokens, consistent with EfficientQAT and QuIP#), we do not observe additional sensitivity compared to prior work. Our appendix also shows robustness to calibration size and sequence length. In addition, similar to QuIP-style methods, more robust alternatives such as LDLQ can be integrated when needed.
>
> ------
>
> **W3: Deployment practicality and hardware support**
>
> We would like to emphasize that LiftQuant does **not require fractional-bit hardware support**. Although the storage bit-width is fractional (e.g., 2.4-bit), the computation only involves a lightweight FP16 linear transform followed by a standard INT1-FP16 GEMV. In practice, deployment is straightforward—we replace Linear layers with the following implementation:
>
> ```
> class LiftUQLinear(nn.Module):
>
>     def __init__(self, in_features, out_features, bias=False, matmul=None, ids=None):
>         super().__init__()
>         # ...
>
>     def forward(self, x):
>         bsz, lens, dim = x.shape
>         x = x.view(-1, self.N, self.M)
>         x = (self.P1 @ (x * self.s1) @ self.P2) * self.s2  # whitening transform (low overhead)
>         x = x.view(bsz, lens, -1)
>         return (torch.ops.mylib.bitblas_dispatch(x, self.weight, self.kernel_ids) * 2 - x.sum(dim=-1, keepdim=True)) * self.scale
> ```
>
> Here, `bitblas_dispatch` is a registered BitBLAS operator compatible with `torch.compile`. The `*2 - x.sum(...)`adjustment converts UINT1-FP16 GEMV into INT1-FP16 GEMV. In practice, we achieve efficient deployment using only standard libraries. While we observed minor compatibility issues between BitBLAS and `torch.compile` for certain tensor shapes, these can be resolved with slight structural adjustments, and we are also preparing a custom INT1-FP16 kernel as a fallback. All reported results use the BitBLAS implementation.
>
> Under the 2.4-bit configuration on RTX 4090D, Llama-2-70B achieves 25.5 tokens/s, corresponding to an effective memory bandwidth utilization exceeding 550 GB/s, indicating efficient hardware utilization despite fractional storage precision. Therefore, the fractional bit-width in LiftQuant is a **memory-level abstraction**, and deployment remains fully compatible with standard GPU operator stacks.
>
> We thank you again for your insightful feedback and will clarify these points in the final version.

---

> > ### Author Rebuttal · Reviewer_nSVV · 2026-04-03
> >
> > Thank you for the clarifications; while helpful, my concerns regarding scalability, calibration robustness, and practical deployment are not fully resolved, so I will maintain my original score.

---

> > > ### Author Response · Authors · 2026-04-07
> > >
> > > Thank you for your candid feedback. We completely agree with your assessment and apologize that our current evaluation could not fully resolve these concerns.
> > >
> > > Rigorously proving scalability and calibration robustness indeed requires much larger-scale dataset ablations and fine-grained system-level benchmarks. We strongly agree that such validation is absolutely crucial for industrial-grade applications. We will dedicate our future work to addressing these exact challenges.
> > >
> > > Thank you again for your high standards and valuable insights!

---

### Official Review · Reviewer_VEV3 · 2026-03-10

**Soundness:** 4
**Presentation:** 4
**Significance:** 3
**Originality:** 3
**Overall Recommendation:** 5
**Confidence:** 3

**Summary:**

LiftQuant proposes a quantization method for use with Large Language Models, where the effective quantization bits can be a non-integer fraction, instead of being constrained to integer bit-width, as with many previous works. This has the benefit of extracting more model performance on hardware where more traditional quantization would have to settle for a lower integer bit-width, since the next integer cannot fit.
The proposed quantization works by “lifting” a low-dimensioned full-precision vector into a higher dimension, but quantized to 1 bit.
The proposed methodology includes solutions to ensure the quantization produces lower errors, accounting for the potentially non-gaussian and intercorrelated nature of the starting weights, and includes several rounds of training to calibrate the different quantization parameters.
Resulting performances on different model architectures and different tasks are presented, in which LiftQuant outperforms or is competitive with previous state-of-the-art methods.

**Compliance With Llm Reviewing Policy:**

Affirmed.

**Key Questions For Authors:**

Training seems to be mentioned in 3 distinct steps in the presented quantization methodology:
1.	The mapping matrix M, trained on Gaussian-random weights
2.	W_q and T*, trained group-wise with a calibration dataset
3.	End-to-end fine-tuning
How long does it take to perform each of these training steps?

Some variables used seem undefined, could you clarify them?
1.	Y in section 3.1
2.	L in equation 3
3.	O, C, and I in Section 3.3
As I understand it, LLM’s weights are composed of typically very long vectors, on the order of hundreds or thousands of elements. When using for example LQ-16/8 where d=8, how are you handling weights vectors that are longer than d?
In Figure 4, some of the interpolated lines between data points seem irregular… In particular, the top-middle figure seem to have a section where the interpolated line is significantly above the 2 closed datapoints at 13-3.0 and 70-2.0, inflating the expected performance. Could you explain that?
How does the inference speed/throughput compare with UQ, or full-precision?

**Limitations:**

yes

**Strengths And Weaknesses:**

Soundness: This paper provides a theoretically and experimentally sound method, able to outperform previous techniques within given hardware limitations.
Presentation: It is overall well presented with clear wording and logical flow, providing helpful figures, especially to visualize the dimensional lift-then-project effect. However, a few variables used in equations are not explicitly defined.
Significance: This method allows extracting more performance out of any given deployment hardware, which may make it attractive in many use cases. However, inference speed when using this method is not compared against more traditional quantization with less overhead: perhaps inference is generally faster using UQ, and that could be more valuable than incremental performance gains in some applications.
Originality: Within the field of neural network compression, it is rare to see techniques that intentionally increase the dimensionality of the model in order to help reduce its size.

---

> ### Author Rebuttal · Authors · 2026-03-31
>
> We sincerely thank you for your positive evaluation and insightful questions. We are glad that you find the method sound, well-presented, and practically meaningful.
>
> -----
> **W1: Inference speed vs UQ / VQ**
>
> Compared to UQ, our method introduces an additional lightweight linear transformation, so a moderate overhead is expected. For example, on Llama-2-70B (2-bit, RTX 4090D), UQ achieves 36.1 tokens/s, while LiftQuant reaches 31.3 tokens/s. However, UQ methods typically suffer from substantial accuracy degradation in the 2–3 bit regime, whereas LiftQuant maintains VQ-level accuracy. At the same time, LiftQuant is significantly more efficient than vector-quantization-based methods, outperforming QTIP (24.5 tokens/s) and VPTQ (9.7 tokens/s reported on A100).
> This demonstrates that LiftQuant occupies a favorable middle ground, achieving a strong balance between accuracy and efficiency in ultra-low-bit settings.
>
> -----
> **Q1: Training cost of the three stages**
>
> We clarify that all training stages are performed offline and the cost is modest compared to model pretraining. Specifically, training the mapping matrix M takes about 30 minutes on an A100 (using 10 random initializations and selecting the best); this is a one-time cost shared across all models. The block-wise optimization ($W_q$, $T*$) takes approximately 12 minutes per block for a 7B model. Finally, the end-to-end fine-tuning takes about 1 hour and 45 minutes on a single A100, or 1 hour and 10 minutes on an H800.
>
> -----
>
> **Q2: Undefined variables**
>
> We will clarify in the revision that $y$ in Section 3.1 should be $w$, L in Eq.(3) denotes the loss function, and OC and IC denote the output and input channels of a linear layer, respectively.
>
> ----
>
> **Q3: Handling long weight vectors**
>
> We handle long weight vectors using block-wise quantization. For example, if d = 16 and the weight dimension is 4096, we divide it into 256 blocks of size 16 and quantize each block independently. This is consistent with standard practices in vector and group quantization.
>
> -----
>
> **Q4: Irregular interpolation in Figure 4**
>
> The envelope curves are generated using quadratic spline interpolation, which may introduce artificial extrema between data points, as in the example you mentioned. We would like to clarify that the interpolated curves are only for visualization, and all conclusions are based on the actual evaluated data points (the blue markers). We will revise the figure to avoid potential confusion.
>
> We thank you again for your helpful feedback. We will incorporate these clarifications and improvements in the final version.

---

> > ### Author Rebuttal · Reviewer_VEV3 · 2026-04-03
> >
> > the authors have clearly answered to my raised questions.

---

### Official Review · Reviewer_RAFL · 2026-03-13

**Soundness:** 3
**Presentation:** 3
**Significance:** 3
**Originality:** 2
**Overall Recommendation:** 4
**Confidence:** 3

**Summary:**

This paper studies LLM compression under GPU memory constraints and focuses on the limitation that most existing quantization methods only support integer bit-widths, while hardware resources are continuous. This mismatch may lead to inefficient deployment.
To address this issue, the authors propose LiftQuant, which enables continuous bit-width quantization. The method maps weights to a higher-dimensional space, encodes them using binary values, and then projects them back to approximate the original weights. By adjusting the dimensional ratio between the lifted representation and the weight block, the effective bit-width can vary continuously.
The framework also includes whitening transformation, block-level optimization, and fine-tuning. Experiments are conducted on multiple LLMs including Llama and Qwen. The results show competitive performance and improved performance–memory trade-offs when using fractional bit-widths.

**Compliance With Llm Reviewing Policy:**

Affirmed.

**Key Questions For Authors:**

- The proposed representation uses binary codes with a linear projection to reconstruct weights. From this perspective, it appears related to binary quantization. Could the authors clarify the conceptual difference between LiftQuant and standard binary quantization methods?
- Are the quantization parameters and bit-width configurations learned through training or optimization? How sensitive are these learned settings to different models or tasks?
- It would also be helpful to clarify the practical implementation of the method. In particular, how are the hardware operators or kernels designed for this representation during inference? Since the method relies on binary codes with linear projection and supports fractional bit-widths, more discussion on operator design and hardware support would help better understand its practicality.

**Limitations:**

The method can still be interpreted as a binary-code representation with a linear projection, so its difference from binary-style quantization could be clarified further. In addition, the pipeline includes several components such as representation learning, whitening, and optimization, which may introduce extra complexity and may require specialized implementation for deployment.

**Strengths And Weaknesses:**

Strengths
- The paper addresses a practical deployment issue by highlighting the mismatch between discrete quantization bit-widths and continuous hardware memory budgets. The idea of continuous bit-width quantization is practically motivated.
- The method design is relatively complete. Besides the core representation, the framework includes whitening transformation, block optimization, and fine-tuning to improve stability and performance.
- Experiments cover multiple LLM families and model sizes and report perplexity, downstream tasks, inference throughput, and ablations.

Weaknesses
- The representation appears closely related to binary quantization. The method uses high-dimensional binary codes followed by a linear projection, which conceptually resembles binary quantization schemes. It would be helpful to clarify what distinguishes this formulation from standard binary quantization and what advantages it provides.
- The pipeline includes several stages such as representation learning, whitening, optimization, and fine-tuning. Compared with simpler quantization approaches, this may increase implementation complexity.
- The robustness and system-level practicality could be further validated. Since the quantization parameters and bit-width configurations appear to be obtained through training or optimization, it is unclear how well these settings transfer across different models. In addition, the paper does not discuss hardware operator or kernel design for the proposed representation, which may be important for practical deployment.

---

> ### Author Rebuttal · Authors · 2026-03-31
>
> **Dear Reviewer RAFL,**
>
> We sincerely thank you for your thoughtful review and constructive feedback. We are glad that you find the problem practically motivated and the evaluation comprehensive.
>
> ---
> **W1 / Q1: Relation to binary quantization**
>
> While both LiftQuant and binary quantization use binary representations, they are fundamentally different paradigms.
>
> Existing binary coding methods (e.g., BCQ) construct a mapping between a floating-point scalar and multiple bits, which is still **scalar quantization**—i.e., nonlinearity is modeled independently per dimension.
>
> In contrast, LiftQuant constructs a mapping from a **D-dimensional binary vector to a d-dimensional floating-point vector** via linear projection, and relies on nearest-neighbor search for encoding. This leads to two fundamental differences:
>
> - **(1) High-dimensional vector quantization vs. scalar quantization**
>   LiftQuant induces a structured, non-uniform **vector codebook** of size 2^D (implicitly), enabling **cross-dimensional nonlinear coupling**. This is fundamentally different from binary quantization, which operates independently per dimension.
> - **(2) Continuous bit-width control**
>   In LiftQuant, the effective bit-width is D/d, which can be tuned continuously (e.g., 2.0 → 2.4 → 2.5). Binary quantization does not provide such flexibility.
>
> Therefore, LiftQuant is better understood as a **hybrid quantization framework** that integrates binary parameterization, a 1-bit uniform decoding pipeline, and vector-quantization-level expressiveness via high-dimensional projection.
>
> As such, it does not belong to existing categories such as BCQ, UQ, or VQ, but instead defines a distinct quantization paradigm, which we denote as LiftQuant (LQ).
>
> ----
> **W2: Pipeline complexity**
>
> We agree that our pipeline includes multiple components. However, this is common in **ultra-low-bit (≈2-bit) quantization**, where simple methods (e.g., GPTQ, Quarot) typically fail to achieve usable accuracy.
>
> State-of-the-art methods such as QuIP# and AQLM also rely on multi-stage pipelines or additional optimization. In this context, LiftQuant follows a similar design philosophy to achieve competitive accuracy.
>
> Importantly, this complexity is confined to offline quantization. At inference time, the computation is highly simplified into a single unified operator, without lookup tables or heterogeneous kernels.
>
> ----
>
> **W2 / Q2: Parameter learning and robustness**
>
> Regarding parameter learning and transferability, we clarify the following.
> - **Bit-width** is manually specified via the ratio D/d.
> - Other quantization parameters (e.g., transformation matrices, scales) are optimized following standard practices in EfficientQAT and QuIP#.
>
> In practice, we tune hyperparameters on Llama-2 (7B/70B) using a few blocks (based on MSE). Then we **reuse the same configuration across other models**(Llama-3, Qwen2.5, Qwen3), with only minor adjustments when instability occurs in rare layers.
>
> ------
>
> **Q3: Practical implementation and hardware support**
>
> We would like to emphasize that LiftQuant does **not require fractional-bit hardware support**. Although the storage bit-width is fractional (e.g., 2.4-bit), the computation only involves a lightweight FP16 linear transform followed by a standard INT1-FP16 GEMV. In practice, deployment is straightforward—we replace Linear layers with the following implementation:
>
> ```
> class LiftUQLinear(nn.Module):
>
>     def __init__(self, in_features, out_features, bias=False, matmul=None, ids=None):
>         super().__init__()
>         # ...
>
>     def forward(self, x):
>         bsz, lens, dim = x.shape
>         x = x.view(-1, self.N, self.M)
>         x = (self.P1 @ (x * self.s1) @ self.P2) * self.s2  # whitening transform (low overhead)
>         x = x.view(bsz, lens, -1)
>         return (torch.ops.mylib.bitblas_dispatch(x, self.weight, self.kernel_ids) * 2 - x.sum(dim=-1, keepdim=True)) * self.scale
> ```
>
> Here, `bitblas_dispatch` is a registered BitBLAS operator compatible with `torch.compile`. The `*2 - x.sum(...)`adjustment converts UINT1-FP16 GEMV into INT1-FP16 GEMV. We achieve efficient deployment using only standard libraries. While we observed minor compatibility issues between BitBLAS and `torch.compile` for certain tensor shapes, these can be resolved with slight structural adjustments, and we are also preparing a custom INT1-FP16 kernel as a fallback.
>
> Under the 2-bit configuration on RTX 4090D, Llama-2-70B achieves 31.3 tokens/s, corresponding to an effective memory bandwidth utilization exceeding 568 GB/s, indicating efficient hardware utilization despite fractional storage precision. Therefore, the fractional bit-width in LiftQuant is a **memory-level abstraction**, and deployment remains fully compatible with standard GPU operator stacks.
>
>
>
> We thank you again for your valuable feedback. We will clarify these points in the final version to better highlight the distinction from binary quantization and the practical deployment benefits.

---

> > ### Author Rebuttal · Reviewer_RAFL · 2026-04-07
> >
> > Thank authors for the effort on their rebuttal. My question is partially resolved. I decide to keep my score.

---

### Official Review · Reviewer_9T4Q · 2026-03-18

**Soundness:** 3
**Presentation:** 3
**Significance:** 4
**Originality:** 4
**Overall Recommendation:** 5
**Confidence:** 4

**Summary:**

The authors propose LiftQuant, a weight-only PTQ method with continuous bit-width control for true Pareto-optimal deployment. It uses a "lift-then-project" mechanism by projecting a simple 1-bit lattice from a tunable $D$-dimensional "lifted" space. LiftQuant leverages previously unused VRAM on GPU for true Pareto-optimal deployment, and is also hardware-friendly.

**Compliance With Llm Reviewing Policy:**

Affirmed.

**Final Justification:**

This paper raises an interesting and practical question (continuous bit-width control) and provides a satisfactory answer with LiftQuant, which (i) is fast with an easy-to-implement CUDA kernel, (ii) has good perplexity and accuracy, and (iii) can exploit dormant VRAM space on GPUs. This is a solid accept to me.

**Key Questions For Authors:**

1. Around line 110, can you make "a corollary of the Central Limit Theorem (CLT) " more explicit with a formal theorem?
1. Do we need the optimization in Equation (1) against a standard Gaussian distribution? I feel like we don't because (i) asymptotic normality means the initial/random $M$ shouldn't be too bad, and (ii) $M$ will be fused in $T^\ast = M T^{-1}$ and everything will be tuned with gradient-based optimization in the end anyway.
1. What theoretical properties do we know about the pseudoinverse-based "Accelerated Nearest-Neighbor Search" heuristic you proposed in subsection 3.1?
1. For $T$ in Equation (2), why don't you use the randomized Hadamard in QuIP#, which has better concentration properties and is faster to apply during dequantization? You initialized $P_1$ and $P_2$ to orthogonal matrices like Hadamard, but still adopted a Kronecker product design. Is it for differentiablility so that you can do finetuning?
1. Ideally, Table 4 should contain FP16 inference speed.
1. The following questions are for my own curiosity, so feel free to skip them:
    1. Can you still have good accuracy if you make the mapping matrix $M$ a random binary matrix? By asymptotic normality, a binarized random $M$ should still give you approximately Gaussian codebook, and the advantage is that you can do UINT1-UINT1 GEMV instead of UINT1-FP16 GEMV. In practice, you don't even need to store/fuse $M$; just store a seed and generate the bits on-the-fly with a PRNG.
    1. For bit width 4 and beyond, can you sample a reasonably-sized subset of the search space to deal with the exponential growth? The ideal is similar to Shannon's random coding argument.
1. Editorial:
    1. Typo in section name "1. Introdoction".
    1. Typo in line 245 "we parameterize the layer-wise whitening transform $D$ in a decomposed form", where $D$ should be $T$.
    1. "The Inflexibility of Integer Bit-Widths." should be a subsection around line 90?

**Limitations:**

Yes. See "Discussion on High-Bit Regimes", "Impact Statement", and Appendix A.

**Strengths And Weaknesses:**

**Strengths**
1. LiftQuant addresses the "deployment gap" where $k$-bit models are not good enough, but $(k+1)$-bit models are too large.
1. Empirically, LiftQuant has both good accuracy and inference speed.
    1. The experiments are extensive, ranging over multiple architectures (Llama-2/3, Qwen-2.5/3), size (3B to 70B), and strong baselines like QTIP.
1. The decoder is very simply and doesn't require specialized optimization, only relying on UINT1-FP16 GEMV operations.
1. The "lift-then-project" mechanism is well-motivated by CLT.
1. The "lift-then-project" mechanism, $T^\ast = M T^{-1}$ fusion, and fully differentiable architecture is novel.
1. The paper includes ablation studies to dissect the contributions of individual design components.
1. The paper is well-written, with an easy-to-follow narrative and good illustrations like Figure 1 and Figure 2.

**Weaknesses**
1. As the authors have acknowledge themselves in subsection 3.1 paragraph "Discussion on High-Bit Regimes", their method doesn't scale beyond 4-bit due to the exponentially growing search space. The exponential growth also prevents them from doing Hessian-based adaptive rounding.

---

> ### Author Rebuttal · Authors · 2026-03-31
>
> **Dear Reviewer 9T4Q,**
>
> We sincerely thank you for your positive assessment of LiftQuant. We deeply appreciate your meticulous reading, the recognition of our novelty and hardware-friendly design, and your highly insightful questions.
>
> ---
> **Q1. Formalizing the CLT Corollary**
>
> To make our theoretical motivation explicit, we will add the following formal statement to Section 3.1, based on the theoretical framework of Diaconis & Freedman (1984):
>
> Let $X \in [-1,1]^D$ be uniformly distributed over the hypercube, and let $M \in R^{d \times D}$ be a projection matrix. As $D → ∞$ and $D>d$, for almost all $M$, the distribution of $MX$ converges **weakly in probability** to a Gaussian distribution $N(0, \Sigma_d)$.
>
> This result formalizes our intuition that projecting a high-dimensional 1-bit lattice produces an approximately Gaussian codebook in the target space.
>
> ---
> **Q2. Is optimizing $M$ in Eq.(1) necessary?**
>
> While asymptotic normality suggests a random M should be reasonable, in practice we operate in a **finite-dimensional regime** (e.g., D = 16 or 32). In this setting, we empirically observe that random $M$ leads to noticeably worse quantization error.
>
> In contrast, optimizing $M$ is both stable and efficient: we find that within ~1000 gradient steps, $M$ consistently converges to the reported Gaussian MSE. Therefore, this step is important to bridge the gap between asymptotic theory and practical low-dimensional deployment.
>
> ---
> **Q3. Theoretical understanding of the pseudoinverse heuristic**
>
> We agree that a formal theoretical guarantee is challenging. While a formal guarantee is difficult, empirically for a 20/10 projection under a Gaussian source, exact nearest-neighbor search yields an MSE of 0.0875, whereas the pseudoinverse heuristic achieves 0.0899 (≈3% gap), indicating that it remains very close to optimal in practice.
>
> ---
> ****Q4. Why Kronecker instead of Hadamard?****
>
> Your intuition regarding differentiability is exactly correct. We did not perform an ablation against the randomized Hadamard transform because our primary design goal for $T$ was to maintain a fully differentiable architecture.
>
> Since standard Hadamard matrices can be recursively constructed via Kronecker products of $2 \times 2$ base matrices, our Kronecker parameterization is theoretically capable of expressing Hadamard-like transforms while remaining fully differentiable. We intentionally adopted this design to pave the way for future Quantization-Aware Training (QAT) experiments, where end-to-end gradient flow through the transformation matrix is essential.
>
> ---
> **Q5. FP16 Inference Speed in Table 4**
>
> The FP16 baseline for the 70B model was omitted because its memory footprint (>130GB VRAM) strictly exceeds the capacity of a single RTX 4090D (24GB), rendering single-GPU execution infeasible.
>
> To provide a fair baseline, we have benchmarked the Llama-2-7B model. As shown in the table below, LiftQuant achieves nearly a 5x speedup. We will update these results in the revised version.
>
> | Model      | Precision       | Throughput (Batch Size 1) |
> | :--------- | :-------------- | :------------------------ |
> | Llama-2-7B | FP16            | 39.2 tokens/s             |
> | Llama-2-7B | LiftQuant 2-bit | 184.4 tokens/s            |
>
> ---
>
> **Q6. Curiosity questions**
>
> - **Binary M:** This is a very interesting and insightful suggestion. But we found a fundamental limitation of binary M: it can cause **collisions in the codebook mapping**, reducing representational capacity. For example, when $D=4$, $d=1$, and $M = [1, 1, -1, 1]$, two different binary codes
>   $W_q = [1, 1, 1, 1]$ and $W_q = [1, 1, -1, -1]$
>   both map to the same projected value $W = MW_q= 2$.
>
>   This collision reduces the effective number of quantization points and breaks the high-dimensional non-uniform codebook property. We observed similar issues when experimenting with Hadamard-initialized M. Therefore, while appealing from a hardware perspective, we believe binary $M$ is fundamentally incompatible with preserving the expressiveness of LiftQuant.
>
> - **Sampling search space for ≥4-bit:** We agree this is a promising direction. As you suggested, sampling a subset of the search space could mitigate the exponential complexity.
>
>   In fact, approaches such as additive codebooks (e.g., AQLM), where
>   $W = M_1 W_{q,1} + M_2 W_{q,2}$, provide a practical way to approximate larger codebooks while keeping search tractable.
>
>   For larger-scale $M$ (to match or surpass methods like QTIP at the same bit-width), we are currently exploring structured constructions of $M$ together with compatible nearest-neighbor search variants to control complexity while preserving coding efficiency.
>
> ---
>
> **Q7. Editorial Corrections**
> We will correct typos in the revised version.
>
> We hope our responses address your concerns, and we thank you again for your constructive feedback and support of our work!

---

> > ### Author Rebuttal · Reviewer_9T4Q · 2026-04-03
> >
> > I have some follow-up questions:
> > 1. What's the entire LiftQuant pipeline? I'm a little confused because Equation (1) optimizes the projection matrix $M$, Equation (3) optimizes the whitening transformation $T$, and the block-wise fine-tuning in Equation (5) further optimizes $T^\ast = MT^{-1}$ directly. Are $M$ and $T$ optimized by Equation (1) and (3) only serving as initialization for $W_q$ and $T^\ast$? I was planning to read the source code, but the repo link https://anonymous.4open.science/r/LiftQuant-DA32 has expired.
> > 1. What's the VRAM usage and execution time for each stage of LiftQuant, e.g., optimizing $M$, optimizing $T$, finding $W_q$, block-wise fine-tuning (optimizing $W_q$ and $T^\ast$), and end-to-end fine-tuning?
> > 1. What's the accuracy of the quantized model if we ablate block-wise fine-tuning? Table 5 only ablates end-to-end finetuning, but not block-wise fine-tuning.
> >
> > I totally understand that you probably don't have the time to answer all of my questions, and that's fine. I'll argue for the acceptance of this paper anyway.

---

> > > ### Author Response · Authors · 2026-04-07
> > >
> > > **Dear Reviewer 9T4Q,**
> > >
> > > Thank you so much for your continued engagement! We deeply appreciate your time and effort.
> > >
> > > First, we sincerely apologize for the expired anonymous repository link https://anonymous.4open.science/r/LiftQuant-DA32 (it had a default 2-month limit). We have just extended its validity for another week so you can access it. Please note that the current codebase is a working research version; we will thoroughly polish and release the final version upon acceptance.
> > >
> > > Here are the detailed answers to your follow-up questions:
> > >
> > > ----------
> > > **1. The Entire LiftQuant Pipeline**
> > >
> > > Your understanding is exactly correct: the $M$ and $T$ optimized in Eq.(1) and Eq.(3) essentially serve as high-quality initializations for the fused operator $T^*$ and the binary weights $W_q$. The complete pipeline is as follows:
> > >
> > >  - Preparation: Train the mapping matrix $M$ for the target bit-width (Eq. 1).
> > >
> > >  - Step 1 (Whitening): For each block, train the whitening transform $T$ using an INT2 STE quantizer to reshape the weight distribution (Eq. 3).
> > >
> > >  - Step 2 (Search & Fusion): Perform nearest-neighbor search to map the floating-point weights into the lifted 1-bit space (finding $W_q$). At this point, $M$ and $T$ are mathematically merged into the fused operator $T^*$.
> > >
> > >  - Step 3 (Block-wise FT): Jointly fine-tune $W_q$ and $T^*$ to minimize the output MSE against the full-precision block (Eq. 5).
> > >
> > >  - Step 4 (Global E2E FT): After all blocks are processed, we perform a global end-to-end fine-tuning on the continuous parameters (scales, transforms). Because $W_q$ is frozen and bit-packed at this stage, the memory footprint is drastically reduced
> > > -------
> > > **2. VRAM Usage and Execution Time**
> > >
> > > For a massive model like Llama-3-70B, the peak VRAM consumption during the entire pipeline is approximately 76 GB. This allows the process to fit comfortably on a single 80GB A100/H800 GPU.
> > >
> > > The total execution time on an H800 is roughly 100 GPU hours, broken down as follows:
> > >  - Per-block optimization: \~15 mins for whitening + 10\~20 mins for nearest-neighbor search (depending on the shape of M) + \~30 mins for block-wise FT.
> > >  - Total for 80 blocks: 80~90 GPU hours. (Note: Since blocks are processed independently, this stage is trivially parallelized across multiple GPUs to reduce wall-clock time).
> > >  - Global E2E FT: \~10 hours.
> > >
> > > -------
> > > **3. Ablation on Block-wise Fine-Tuning**
> > >
> > > If we completely ablate the block-wise fine-tuning, the accuracy degrades severely. Immediately after the Round-to-Nearest (RTN) search in the high-dimensional space, the block's output MSE is initially very high.
> > >
> > > However, the optimization landscape is highly favorable: within just a few gradient steps of block-wise FT, the MSE drops rapidly. For most blocks, the MSE stabilizes after only 10% of our default training steps (approx. 1M tokens). We maintain the 8M token setting strictly to ensure convergence for a few exceptionally difficult layers. Because the initial RTN error is large, block-wise FT is an indispensable step in our pipeline—a practice that is also standard and necessary in other state-of-the-art methods like QTIP and EfficientQAT.
> > >
> > > Thank you again for your fantastic support and insightful questions! We will make sure to clarify the pipeline and computational costs explicitly in the final manuscript.

---

### Decision · Program_Chairs · 2026-04-30

**Decision:**

Accept (spotlight)

**Comment:**

This paper proposes a learnable vector quantization method to achieve continuous bit-width control (non-integer bits) as well as nonuniform quantization. The hardware-aware compression framing is well-motivated, and the proposed approach is elegant. Reviewers are all positive.